# Effects of Telerehabilitation-Based Structured Home Program on Activity, Participation and Goal Achievement in Preschool Children with Cerebral Palsy: A Triple-Blinded Randomized Controlled Trial

**DOI:** 10.3390/children10030424

**Published:** 2023-02-22

**Authors:** Sinem Asena Sel, Mintaze Kerem Günel, Sabri Erdem, Merve Tunçdemir

**Affiliations:** 1Faculty of Health Sciences, Department of Physiotherapy and Rehabilitation, Antalya Bilim University, Antalya 07190, Turkey; 2Faculty of Physical Therapy and Rehabilitation, Hacettepe University, Ankara 06230, Turkey; 3Faculty of Business, Dokuz Eylul University, İzmir 35390, Turkey

**Keywords:** cerebral palsy, telerehabilitation, physiotherapy, home program, participation

## Abstract

A home program is implemented as an evidence-based mode of delivering services for physiotherapy and rehabilitation. Telerehabilitation is a method applied in physiotherapy modalities for children. This study aims to determine the effectiveness of usual care plus a Telerehabilitation-Based Structured Home Program on preschool children with cerebral palsy (CP) compared to usual care. Forty-three children aged 3–6 years (mean age 4.66 ± 1.08 years) with CP were randomly assigned to the Telerehabilitation-Based Structured Home Program and usual care groups. Their motor function was assessed with the Gross Motor Function Measure (GMFM); performance and satisfaction were evaluated with the Canadian Occupational Performance Measure (COPM); goal achievement was assessed with the Goal Attainment Scale (GAS); and activity and participation were evaluated with Pediatric Evaluation of Disability Inventory (PEDI). Participants were evaluated at baseline, immediately post-intervention (12 weeks) and at follow-up (24 weeks). There was a statistically significant difference between pre- and post-test GMFM, COPM, GAS and PEDI scores in the intervention and control groups (*p* < 0.001). The Telerehabilitation-Based Structured Home Program showed statistically significant changes in activity, participation and goal achievement after 12 weeks of intervention (*p* < 0.001). However, significant results were not obtained in the usual care group. The Telerehabilitation-Based Structured Home Program may be an effective method for preschool children with CP. (Registration number: NCT04807790; no = KA-20124/26.01.2021).

## 1. Introduction

Cerebral palsy (CP) is a condition in which movement development and postural disorders that occur as a result of non-progressive damage to the developing brain cause activity limitations. Motor disorders are often accompanied by sensory, cognition, communication, perception, behavioral disorders and epilepsy [1].

Activity is defined as a specific “task or action” performed by an individual and participation is defined as “participation in a life situation” [2]. Participating in activities strengthens children’s academic performance, shapes their social relationships and enables the development of their identities [3]. When academic performance is considered in terms of social relations and personal identity, participation plays a key role in the child’s development and quality of life [4]. On the other hand, the physical activity of children with CP in the preschool period is less than children with typical development due to motor disorders, and therefore they experience limitations in activity and participation [5,6].

The International Classification of Functioning, Disability and Health-Children and Youth (ICF-CY) examines the child’s basic health-related status in three areas: body structure and function, activity and participation [7]. When children with CP in the preschool period were examined within the framework of ICF-CY, it was determined that they had significant activity and participation limitations due to body functions and structure, environmental factors and personal factors [8]. As a result, physiotherapy and rehabilitation intervention programs should be arranged to increase activity and participation [9].

A home program is defined as “therapeutic activities performed at home by the child with the help of parents to achieve health-related goals” [10]. Using the support and guidance obtained by applying the home program, families build confidence in an area in which they can help their children [11]. According to Novak et al., a home program is an evidence-based mode of delivering services and is used as a guide and support in many physiotherapy modalities [12]. A home program is used together with methods such as virtual reality, modified constrained induced movement therapy, action observation therapy, mirror therapy, bimanual training, strength training, goal-directed therapy and neurodevelopmental therapy [13]. The main goal of the home program is to increase the activity level of children with CP, to ensure that acquired skills are used in daily life, and to prevent secondary problems that can be seen in the long term [10]. Therefore, it is an important approach for pediatric physiotherapists [12].

Telerehabilitation (TR) is the use of electronic information and telecommunications technologies to support long-distance clinical health care, patient and professional health education, public health and health administration [14]. The goals of TR are to provide distance education and counseling, evaluate rehabilitation interventions and monitor their results and carry out a distance-therapeutic intervention program [14]. Studies show that TR can be effective in the field of pediatric physiotherapy and rehabilitation [15]. The TR model, which is family centered, coached by physiotherapists and sets individual goals, can be as effective as face-to-face intervention methods [15]. However, there is no randomized controlled study with a high level of evidence in the field of TR for children with CP.

This study aims to compare motor functions, activity and participation between pre-school children with CP who receive routine physiotherapy and rehabilitation, and children who receive routine physiotherapy and rehabilitation plus a “Telerehabilitation Based Structured Home Program”.

## 2. Materials and Methods

This randomized controlled trial compares usual care (control group) and a usual care + Telerehabilitation-Based Structured Home Program (study group) on activity, participation and goal achievement. This study was planned as a triple-blind study. The therapist implementing the intervention was blinded from the evaluation and analysis of the data. The therapist who made the scoring was blinded from the study and control groups. The researcher who performed the data analysis was also blinded from the study and control groups. The study protocol was approved by the institutional ethics board of Hacettepe University (registration number: NCT04807790; no = KA-20124/26.01.2021). Parents who agreed to participate in the study signed a written consent form. The paper is registered with ClinicalTrials.gov. The clinical trial number for this study is NCT04807790.

### 2.1. Participants

One hundred and twelve children with CP who attended an evaluation in the Hacettepe University Physical Therapy and Rehabilitation Faculty, Cerebral Palsy and Pediatric Rehabilitation Unit were screened for this study. In the power analysis, the error level of α: 0.05 was determined with 80% power and the sample size was determined. A total sample size of 48 children with CP, 24 in each group, was calculated based on a Gross Motor Function Measure (GMFM) from a previous study protocol using G*Power Software v 3.1.9.4, Windows 10, Heinrich Heine Universität Düsseldorf [16]. Forty-eight parents of children accepted participation in the study. The children were then randomized into two groups (Figure 1). The study design has been informed by Consolidated Standards of Reporting Trials Guidelines [17].

The inclusion criteria were (1) age between 3 and 6 years, (2) receiving two sessions of usual care service per week and (3) having family members actively using mobile phones, computers and the internet. The exclusion criteria were (1) any orthopedic surgery related to the lower or upper extremities in the last 6 months, (2) botulinum toxin injection to the upper or lower extremity muscles in the last 6 months and (3) uncontrollable epileptic seizures.

### 2.2. Procedures

Demographic characteristics (i.e., age and sex) and severity of functional impairments (i.e., gross motor function, manual ability and communication) of the 48 preschool children with CP included in the study were obtained by a physiotherapist. Among the 48 participants, 24 were randomized to the study group and 24 to the control group. Equal and random distribution of the sample group to the study groups was provided by the closed/sealed envelope method, in which the envelopes containing the papers on which the names of the children were written were mixed thoroughly and randomly distributed as 24 people in each group (Figure 1). The study group was defined as the Telerehabilitation-Based Structured Home Program and the control group was defined as usual care.

After the randomization process, parents were informed about their allocated intervention in detail then baseline assessment videos were recorded.

### 2.3. Classification Tools

Classification systems were applied to describe the functional levels of participants. The Gross Motor Function Classification System (GMFCS) (ICC = 0.75) [18], Manual Ability Classification System (MACS) (ICC = 0.96) [19] and Communication Function Classification System (CFCS) (ICC = 0.77) [20] were applied by an experienced physiotherapist (SAS).

### 2.4. Outcome Measures

Video recordings of GMFM-66 were taken by the first researcher (SAS). Scoring was then done by a second researcher (MT, MSc) who had 7 years of experience in pediatric rehabilitation and was completely blinded from the scoring and statistical process. PEDI, COPM and GAS scores were made by parents. The first researcher was completely blinded from the scoring and statistical process. Assessment tools were chosen to measure activity, participation and goal achievement.

#### 2.4.1. Gross Motor Function Measure-66 (GMFM-66)

GMFM-66 is a valid clinical scale designed to evaluate the change in the gross motor functions of children with CP. It measures activity according to ICF-CY [21]. The validity and reliability study of GMFM has been proven to be a tool that can be evaluated through video recording [22].

#### 2.4.2. Pediatric Evaluation of Disability Inventory (PEDI)

PEDI has been developed to evaluate activities of daily living. It consists of three main sections under the subdomains of functional skills, caregiver assistance and modifications [23]. In this study, the aim of using PEDI was to determine the activity and participation levels of participants.

#### 2.4.3. Goal Attainment Scale (GAS)

The GAS is used to measure individual progress in rehabilitation. It can determine the extent to which individuals can achieve the determined goals. The therapist-specific GAS is graded between −2 and +2. The initial value of the goals was determined as −1. A value of 0 was reached after the intervention was accepted as goal attainment. The primary outcome was defined as attainment of at least one goal to an expected, or greater than expected, level at T1. This binary interpretation is consistent with recommendations for using the GAS in a pediatric rehabilitation as it allows appropriate statistical analysis and has been shown to identify clinically meaningful improvements. GAS scores were calculated according to the T-total formula [24]. For a goal to be considered successful, the T value must be 50 or higher. The purpose of using the GAS in this study is to measure the rate of goal achievement. Table 1 includes an example of GAS applied to one of the participants.

#### 2.4.4. Canadian Occupational Performance Measure (COPM)

COPM is an individualized scale that measures the change in performance and satisfaction over time, based on an individual’s perception [25]. The COPM is a feasible multidisciplinary outcome measure for pediatric TR [26]. This measurement evaluates the activity performance problems of the individuals with the performance and satisfaction scores they give themselves. The COPM was used to measure the performance obtained as a result of the intervention and the satisfaction level of the parents.

#### 2.4.5. Interventions

After informed consent was obtained, the researcher (SAS: PT, MSc) went to the participants’ homes. During the visit, the home environment of the participant was examined. Evaluation videos of the participants were taken. After the evaluation, the researcher interviewed the parents face to face for a second time. During this meeting, three goals were determined according to COPM and GAS. Participants in the study group were instructed to meet face to face after 12 weeks (post-intervention) and 24 weeks (follow-up). The intervention group received structured home program education.

### 2.5. Telerehabilitation-Based Structured Home Program

The Telerehabilitation-Based Structured Home Program consist of Telerehabilitation and structured home program phases. Twelve hours of therapy were a part of TR (1 h per week × 12) 56 h of therapy were applied at home by parents of children with CP (7 weekdays × 40 min × 12 weeks = 56 h of home practice). At the end of the therapy, 68 h were completed. The settled structured home program was applied according to the daily schedule of the family by parents in the home environment. The therapist and parent met online. Each session of the TR lasted 1 h. Structured home program education was tailored specifically to each child. After the training, parents applied the structured home program at home for 40 min every day. SAS followed families with online meetings and provided regular meetings every week.

The structured home program education was given by SAS, with 7 years of PT experience, according to the children’s and parents’ goals. The PT (SAS) trained the parent face to face for an hour. The education aimed to ensure the family is more involved in the process, to achieve the goals. The content of the structured home program was determined by the family according to the targeted GAS and COPM goals and consisted of goal-oriented activities. Goal-oriented activities aimed to increase activity and participation and were determined specifically for each child. After the training was given, the telerehabilitation phase of the protocol was started. The same PT (SAS) prescribed the home program through WhatsApp video calls per week for 1 h. Online interviews continued for 12 weeks.

Assessments were conducted at Hacettepe University, Physical Therapy and Rehabilitation Faculty, Cerebral Palsy and Pediatric Rehabilitation Unit. The intervention’s TR part was completed through WhatsApp video calls and the structured home program was conducted at the children’s home (Figure 2).

Participants’ home program content was checked by reading diaries and meetings with first researcher who implemented the intervention, an experienced physiotherapist MKG (PT, Ph.D.), and was blinded from study and control groups. 

### 2.6. Usual Care

In Turkey, during routine physiotherapy and rehabilitation implementation, activities are carried out by the physiotherapist according to the functional level of children with CP. Physiotherapists working in special education centers in Turkey conducted usual care. Goals are defined by the therapist only and aimed to improve body function and structures. Approaches such as stretching, splinting and casting are used frequently. In the control group, the total dose of usual care was two sessions per week for 12 weeks 40 min = 16 h + home program.

Parents of children in each group kept a log of usual care therapy including frequency, duration, mode and content. After the intervention, the total home program implementation time was calculated. The control group only received usual care and was evaluated by PT (SAS) for 6 months (at post-intervention and follow-up periods).

### 2.7. Statistical Analysis

The IBM SPSS statistical software 23.0(Chicago, USA) was used for statistical analysis. One-sample Kolmogorov–Smirnov tests were used to evaluate the distribution of variables before test selection. Physical characteristics in the experimental and control groups were compared using the χ^2^ test for categorical variables (sex, functional level). The Mann–Whitney U test was used to analyze the continuous variable (age). Baseline, post-treatment and follow-up were calculated within groups. A one-way ANOVA test (1 group × 3 tests) was used to compare the differences in assessments between treatment scores within groups at baseline and after 12 weeks and follow-up (24 weeks). When overall significance was found, a pairwise post-hoc test with Bonferroni correction was performed based on an adjusted *p*-value of 0.017 (0.05/3; dividing *p*-value to group number). The difference between the groups was determined with two-way repeated-measures mixed ANOVA (2 groups × 3 tests) for normally distributed numerical data. To determine the effect sizes (ES) of the interventions on the measured properties, Cohen’s d formula was applied. The ES was classified as follows: large ES when greater than 0.80, medium ES when 0.50 to 0.80 and small ES when less than 0.50 [27].

## 3. Results

The mean age of 43 children who completed the study was 4.66 ± 1.08 years. According to demographic data, two groups were similar in terms of age, gender and functional levels (GMFCS, MACS, CFCS) (Table 2). Between January 2021 and March 2022, 43 children with CP in the preschool period were enrolled in the study (*n* = 23) and control (*n* = 20) groups. At baseline, the study and control groups were similar in terms of GMFM-66, PEDI, COPM and GAS (*p* > 0.05) (Table 3).

At post-intervention, after 12 weeks in the Telerehabilitation-Based Structured Home Program group within-groups significant differences were found (*p* < 0.001). One-way ANOVA demonstrated that the Telerehabilitation-Based Structured Home Program approaches led to significant improvements in activity, participation and goal achievement (*p* < 0.05). GMFM (F = 10.90, *p* = 0.001), COPM-Performance (F = 7.03, *p* = 0.005), COPM-Satisfaction (F = 7.07, *p* = 0.005), PEDI-Mobility (F = 26.86, *p* < 0.001), PEDI-Self Care (F = 13.87, *p* < 0.001) and GAS (Z = −4.20, *p* < 0.001) results found statistically significant (*p* < 0.05) (Table 3). However, within groups in usual care groups, there was no significant difference (*p* > 0.05) (Table 4).

Between groups, there was a significant difference in GMFM (F = 14.86, *p* < 0.001), COPM-Performance (F = 3.34, *p* = 0.040), COPM-Satisfaction (F = 6.68, *p* = 0.006), PEDI-Mobility (F = 18.90, *p* < 0.001), PEDI-Self Care (F = 23.05, *p* < 0.001) and GAS (Z = −4.20, *p* < 0.001) (Table 5). ESs of PEDI-Mobility (d_T1–T2_ = 1.12, d_T1–T3_ = 1.14) and PEDI-Self Care (d_T1–T2_ = 0.70, d_T1–T3_ = 0.78) results found moderate to large differences at immediate post-intervention and follow-up period. GAS results were found to be very large at the immediate post-intervention and follow-up period (d_T1–T2_ = 4.30, d_T1–T3_ = 4.64). In the study group, a total of 69 goals were determined. The mean score of the goals was 63.60 ± 8.52. Twenty-nine goals were related to gross motor function activities, 20 goals to fine motor function activities and 20 goals to participation. In the control group, a total of 69 goals were determined. The mean score of the goals was 37.94 ± 3.96. Twenty-six goals were gross motor function activities, 18 were fine motor function activities and 18 were participation. The home program implementation dose was significantly different between groups (Z = −5530, *p* < 0.001) (Table 6).

COPM-Performance and COPM-Satisfaction results both showed statistically significant changes and improvements in the Telerehabilitation-Based Structured Home Program group. In addition, between groups, COPM-Performance and COPM-Satisfaction results both demonstrated statistically significant changes (Figure 3). ESs of COPM-Performance results were found to be large in the immediate post-intervention and follow-up period (d_T1–T2_ = 1.46, d_T1–T3_ = 0.80). COPM-Satisfaction showed moderate to large ES results (d_T1–T2_ = 0.60, d_T1–T3_ = 0.35) (Table 4).

## 4. Discussion

This study examined the effect of usual care plus a Telerehabilitation-Based Structured Home Program on motor functions, activity and participation in preschool children with CP and compared it to the control group, which only received usual care. Overall, the study findings showed that the usual care plus the Telerehabilitation-Based Structured Home Program was more effective than usual care. The Telerehabilitation-Based Structured Home Program showed significant improvement in motor function, activity and participation. In particular, the Telerehabilitation-Based Structured Home Program group achieved significant progress in the success of the goals. However, usual care did not show similar improvements.

Participation in home, school and community activities is a primary outcome for preschool-age children with disabilities and their families [28]. Play activities, skill development, active physical recreation and social activities are the most common activities in which preschool children with CP participated [29]. Abu-Dahab et al. determined that preschool children show low participation and activity levels for typically developing children [30]. Novak et al. reported a relationship between goal-directed home program intervention and participation in children with CP aged between 2 and 7 years [10]. Goals in physiotherapy and rehabilitation for children with CP should be determined according to participation and activity; furthermore, intervention programs should be designed specific to this principle [31]. Therefore, goal-oriented, participation and activity-based interventions for preschool children with CP is substantial. In this respect, the results of the study are compatible with the results of other studies. The Telerehabilitation-Based Structured Home Program protocol was developed based on goal-oriented activity and participation, and determined that it was more effective than routine physiotherapy in pairwise group comparisons.

Each study group received usual care in rehabilitation centers. The study group committed to performing the Telerehabilitation-Based Structured Home Program and demonstrated significant progression in activity, participation and motor function. When we compared usual care and the Telerehabilitation-Based Structured Home Program group, the Telerehabilitation-Based Structured Home Program group was more effective in activity, participation and motor function. The results show that usual care alone is not effective and a different modality such as TR is needed. From this point of view, routine physiotherapy ceases to be an effective method due to the poor use of resources. In the current study, motor function outcomes have a linear relationship with activity and participation. Moreover, when goal achievement is successful, activity and participation outcomes are improved. Studies support our results in this regard. Ferre et al. stated that TR with home-based intensive care improves motor function functional goals, activity and participation by means of COPM outcomes [32]. Similar outcomes were reported by Surana et al., stating that home-based intensive care plus TR for lower extremities is effective for motor function and activity.

In post-intervention, all ESs had a large range of Telerehabilitation-Based Structured Home Program (d = 0.70–4.64). The usual care group was low in all ES indexes except GAS (d = 0.07–0.57). The current study hypothesized that Telerehabilitation-Based Structured Home Program is more effective in the post-intervention period than usual care. Group comparisons demonstrated that the Telerehabilitation-Based Structured Home Program was more effective in motor function, activity, and participation. On the contrary, Novak et al. stated that the home program did not affect participation [33]. The reason for the contrast between the study results may be attributed to differences in age groups, different outcome measures, the existence of TR and intervention dosage. Significant difference between usual care and the Telerehabilitation-Based Structured Home Program group maintained during the follow-up period. ESs of motor function, activity and participation during the follow-up period remained. The gains of the Telerehabilitation-Based Structured Home Program as an intervention method continued.

Satisfaction is correlated with parenting and is important in pediatric rehabilitation [34]. In this study, family satisfaction was assessed with COPM. Outcomes of COPM improved in the Telerehabilitation-Based Structured Home Program group but post-intervention results of the ES index were low (d = 0.16–0.35). On the other hand, the satisfaction outcomes of the usual care group were not statistically significant. Based on the results, we think the presence of physiotherapist guidance and consultants through TR during the treatment affects the satisfaction of the parents. Bamm et al. stated that education and counseling are elements of rehabilitation [35]. In usual care, goal setting and therapy are done by the physical therapist only, and education and counseling for parents are limited [9,36]. However, in the Telerehabilitation-Based Structured Home Program protocol, the goals are determined in partnership with the family and the therapist, the home program training is given under the guidance of the therapist and is created in a way that will be implemented by the family. The physiotherapist uses TR for counseling and education during the rehabilitation process. From this point of view, the reason why the Telerehabilitation-Based Structured Home Program is more effective in terms of activity, participation, and goal attainment compared to routine physiotherapy can be understood as family education and counseling.

Goal setting is a fundamental component of decision-making and outcomes in pediatric rehabilitation [9]. In the current study, goals were determined according to children’s activity and participation phrases. GAS outcomes showed significant changes in the Telerehabilitation-Based Structured Home Program group and ES indexes were large (d = 4.30–4.64). The current study hypothesized that the Telerehabilitation-Based Structured Home Program is effective in goal achievement. Novak et al. used GAS to measure goal attainment and found a significant relationship between home program and goal achievement [35]. ES results were very large, just as in the current study. This consistency demonstrates the home program is effective in goal achievement. Post-intervention results of ESs were also large. Goal achievement in the Telerehabilitation-Based Structured Home Program group persisted during the follow-up period. ØstensjØ et al. stated that GAS and COPM should be used in combination to assess activity, participation and goal achievement [37]. GAS and COPM were used together in the current study. Therefore, this study assessed activity, participation and goal achievement in all aspects. Outcomes are significant and promising for clinical use in preschool children with CP.

TR is an effective method in pediatric rehabilitation. Camden et al. stated that a family-centered and goal-directed TR model is as effective as face-to-face intervention methods [15]. The goal-oriented home program created in cooperation with the parents can be implemented effectively without TR support [9]. However, the current study demonstrated that TR is a method that can support the goal-directed home program. TR can be a promoting modality for the home program. Overall, the regular TR method with the structured home program can be used for preschool children with CP to improve motor function, activity and participation.

In the current study, the burden of TR on the family’s quality of life was not examined. Considering that TR requires a certain time and environment, it can be thought that it may affect the quality of life of the family. Future studies should examine the effects of TR on the quality of life of the family. In addition, we considered the rate of implementation of the treatment may decrease after the end of the intervention, due to the high dose of therapy and the need for regular consultation via TR.

Preparing preschool children with CP for school, social life and daily living activities is crucial for their future life direction [28]. At this point, interventions of this period must focus on this principle. The Telerehabilitation-Based Structured Home Program protocol aims to support families and preschool children with CP in a home environment through TR. When the results and purpose are compared, it can be thought that Telerehabilitation-Based Structured Home Program is a goal-directed method that can be applied in the home environment and has a positive effect on activity, participation and motor function. Consequently, regular Telerehabilitation-Based Structured Home Program intervention provided in a home environment, can improve motor function, activity and participation outcomes.

### 4.1. Adverse Events

There were no significant adverse events were reported.

### 4.2. Limitations

Finding a suitable day and time for parents to make video calls created some difficulties in terms of completing the protocol. Since CP is a heterogeneous group, there was a trend to have more severely affected children with CP in the control group. Another limitation of the study was the inability of parents to be blinded from the study.

## 5. Conclusions

In conclusion, preschool children with CP are in a crucial period of their lives in terms of physiotherapy and rehabilitation. For this reason, they need activity, participation and a goal-oriented approach, which are the goals that the family and physiotherapist cooperate with. According to the results of this study, the Telerehabilitation-Based Structured Home Program intervention is at a level that can meet the needs of preschool children with CP.

## Figures and Tables

**Figure 1 children-10-00424-f001:**
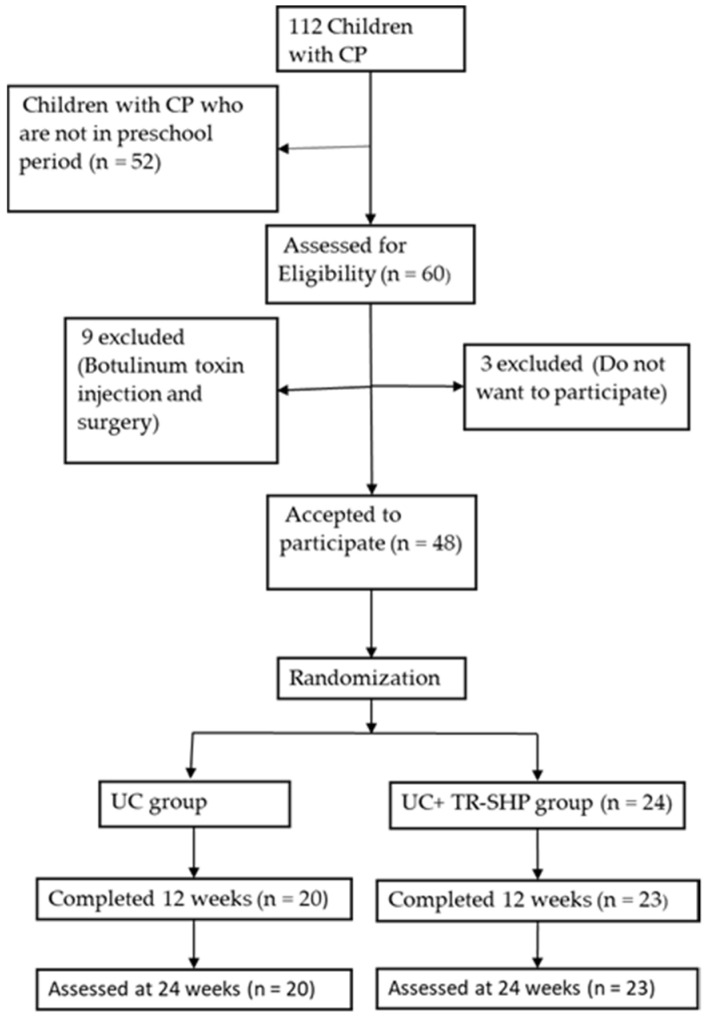
CONSORT flow diagram of patient recruitment and participation throughout the study.

**Figure 2 children-10-00424-f002:**
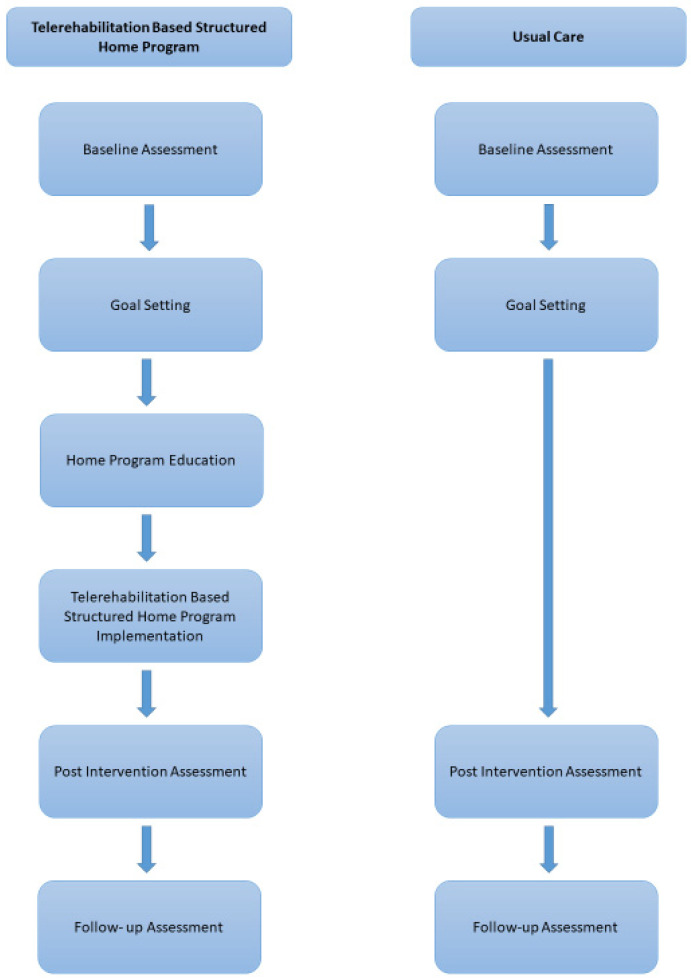
Telerehabilitation-Based Structured Home Program Approach diagram.

**Figure 3 children-10-00424-f003:**
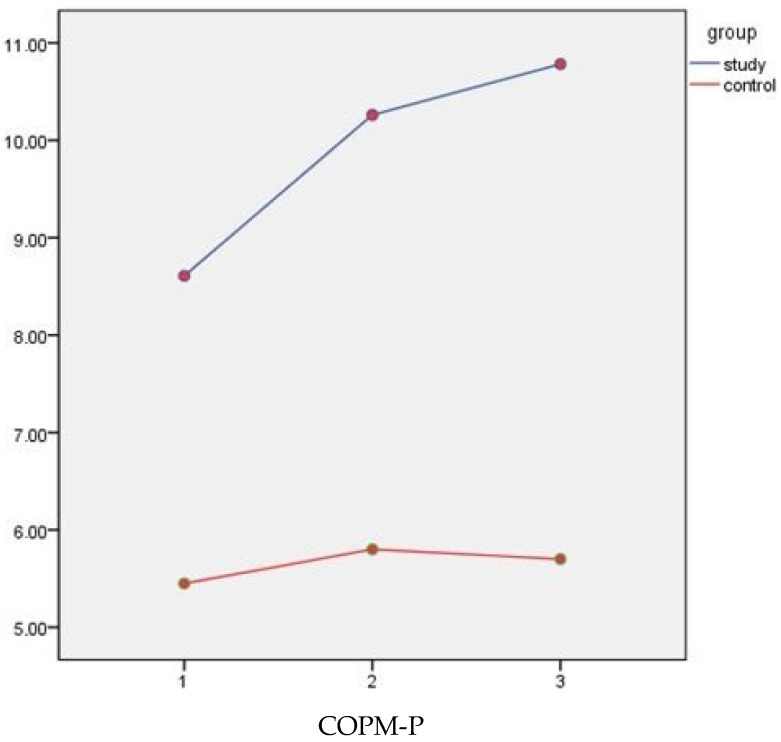
COPM-P and COPM-S graphic of change over time between groups.

**Table 1 children-10-00424-t001:** Goal-attainment scale examples.

Attainment Level	Score	Activity-Based Example	Participation-Based Example
Much less than expected outcome	−2	Cannot use the assistive device with the support of another person.	Cannot ambulate in kindergarten with the support of another person with the assistive device.
Somewhat less than expected outcome	−1	Can use the assistive device with the support of another person.	Can ambulate in kindergarten with the support of someone else with the assistive device.
Expected performance by the end of the measurement period	0	Can use the assistive device with close supervision of their parents.	Can ambulate in kindergarten independently with close supervision of their teacher.
Somewhat more than expected outcome	1	Can use the assistive device independently for 1 m.	Can walk in kindergarten 1 m independently with an assistive device.
Much more than expected outcome	2	Can use the assistive device independently for 5 m	Can move around the kindergarten with an assistive device.

**Table 2 children-10-00424-t002:** Comparison of demographic and physical characteristics of the groups.

		UC + TR-SHP (*n* = 23)		UC (*n* = 20)		Z	*p* *
		Mean ± SD		Mean ± SD			
Age (years)		4.63 ± 1.06		4.70 ± 1.14		−0.272	0.785
		** *n* **	**%**	** *n* **	**%**	**χ^2^**	** *p* **
Gender	Girl	11	47.8	12	40	0.637	0.425
	Boy	12	52.2	8	60		
GMFCS	Level I	6	26.1	3	15	−0.801	0.423
	Level II	2	8.7	2	10		
	Level III	4	17.4	6	30		
	Level IV	6	26.1	1	5		
	Level V	5	21.7	8	40		
MACS	Level I	6	26.1	6	30	−0.188	0.851
	Level II	1	4.3	3	15		
	Level III	6	26.1	3	15		
	Level IV	4	17.4	1	5		
	Level V	6	26.1	7	35		
CFCS	Level I	13	56.5	9	45	−1.125	0.261
	Level II	0	0	0	0		
	Level III	3	13	1	5		
	Level IV	3	13	3	15		
	Level V	4	17.4	7	35		

TR-SHP, Telerehabilitation-Based Structured Home Program; UC, usual care; SD, standard deviation; MACS, Manual Ability Classification System; CFCS, Communication Function Classification System; GMFCS, Gross Motor Function Classification System; Z, Mann–Whitney U test, χ^2^, Chi-squared test, * *p* < 0.05.

**Table 3 children-10-00424-t003:** Initial characteristics of groups determined by measurement.

	UC + TR-SHP (*n* = 23)	UC (*n* = 20)	T	*p* *
	Mean ± SD	Mean ± SD		
GAS	36.36 ± 0.04	36.34 ± 0.05	1.642	0.100
GMFM	49.15 ± 38.53	42.07 ± 37.25	0.793	0.610
COPM-P	8.60 ± 0.84	5.45 ± 2.28	1.845	0.055
COPM-S	8.60 ± 4.05	5.55 ± 0.54	1.970	0.221
PEDI-SC	27.82 ± 25.45	22.45 ± 22.20	0.733	0.154
PEDI-MOB	23.73 ± 24.46	14.35 ± 19.88	1.367	0.031
PEDI-SF	24.73 ± 24.98	25.30 ± 26.41	−0.071	0.306

TR-SHP, Telerehabilitation-Based Structured Home Program; UC, usual care; SD, standard deviation; GAS, Goal Attainment Scale; GMFM-66, Gross Motor Function Measure-66; COPM-P, Canadian Occupational Performance Measure Performance; COPM-S, Canadian Occupational Performance Measure Satisfaction; PEDI-SC, Pediatric Evaluation of Disability Inventory Self Care; PEDI-MOB, Pediatric Evaluation of Disability Inventory Mobilization; PEDI-SF, Pediatric Evaluation of Disability Inventory Social Function; T, Student’s *t*-test; * *p* < 0.05.

**Table 4 children-10-00424-t004:** Motor function, activity and participation change through time according to each group.

	UC + TR-SHP (*n* = 23)					UC (*n* = 20)				
	T1	T2	T3	*p* *	Pairwise Comparison	T1	T2	T3	*p* *	Pairwise Comparison
	X ± SD	X ± SD	X ± SD			X ± SD	X ± SD	X ± SD		
GMFM	49.15 ± 38.53	53.15 ± 3.25	53.20 ± 38.17	**0.001 ***	T1–T2 T1–T3	42.07 ± 37.25	42.34 ± 37.14	42.60 ± 37.29	0.053	-
PEDI-SC	27.82 ± 25.45	30.69 ± 25.50	30.73 ± 25.58	**0.000 ***	T1–T2 T1–T3	22.45 ± 22.20	22.50 ± 22.28	22.50 ± 22.28	0.330	-
PEDI-MOB	23.73 ± 24.46	25.60 ± 24.36	25.65 ± 24.33	**0.000 ***	T1–T2 T1–T3	14.35 ± 19.88	14.45 ± 19.81	14.55 ± 19.89	0.174	-
PEDI-SF	24.73 ± 24.98	24.95 ± 24.96	24.95 ± 24.96	0.135	-	25.30 ± 26.41	25.30 ± 26.41	25.30 ± 26.41	1.00	-
COPM-P	8.60 ± 0.84	10.26 ± 0.91	10.78 ± 0.99	**0.005 ***	T1–T3	5.45 ± 2.28	5.80 ± 2.48	5.70 ± 3.22	0.609	-
COPM-S	8.60 ± 4.05	10.26 ± 4.39	10.78 ± 4.76	**0.005 ***	T1–T3	5.55 ± 0.54	5.15 ± 0.55	5.45 ± 0.72	0.232	-
GAS	36.36 ± 0.04	62.28 ± 8.52	60.14 ± 8.82	**0.000 ***	T1–T2 T1–T3	36.34 ± 0.05	37.94 ± 3.96	37.26 ± 2.78	**0.24**	-

TR-SHP, Telerehabilitation-Based Structured Home Program; UC, usual care; X, mean; SD, standard deviation; GAS, Goal Attainment Scale; GMFM-66, Gross Motor Function Measure-66; COPM-P, Canadian Occupational Performance Measure Performance; COPM-S, Canadian Occupational Performance Measure Satisfaction; PEDI-SC, Pediatric Evaluation of Disability Inventory Self Care; PEDI-MOB, Pediatric Evaluation of Disability Inventory Mobilization; PEDI-SF, Pediatric Evaluation of Disability Inventory Social Function; T, testing session; * one-way repeated measures ANOVA; * pairwise comparisons with Bonferroni correction; note: For the pairwise comparisons, the mean difference is significant, at 0.017 level.

**Table 5 children-10-00424-t005:** Comparison of TR-SHP and UC motor function, activity and participation outcomes.

	T	UC + TR-SHP (*n* = 23)	Effect Size (*d*)		UC (20)	Effect Size (*d*)		*p* *
			T1–T2	T1–T3		T1–T2	T1–T3	
		X ± SD			X ± SD			F/*p*
**GMFM**	**T1**	49.15 ± 38.53	1.04	1.05	42.07 ± 37.25	0.07	0.14	**14.86/0.000 ***
	**T2**	53.15 ± 38.25			42.34 ± 37.14			
	**T3**	53.20 ± 38.17			42.60 ± 37.29			
**PEDI-SC**	**T1**	27.82 ± 25.45	1.12	1.14	22.45 ± 22.20	0.22	0.22	**23.05/0.000 ***
	**T2**	30.69 ± 25.50			22.50 ± 22.28			
	**T3**	30.73 ± 25.58			22.50 ± 22.28			
**PEDI-MOB**	**T1**	23.73 ± 24.46	0.70	0.78	14.35 ± 19.88	0.50	0.50	**18.90/0.000 ***
	**T2**	25.60 ± 24.36			14.45 ± 19.81			
	**T3**	25.65 ± 24.33			14.55 ± 19.89			
**PEDI-SF**	**T1**	24.73 ± 24.98	0.08	0.08	25.30 ± 26.41	0.00	0.00	1.74/0.194
	**T2**	24.95 ± 24.96			25.30 ± 26.41			
	**T3**	24.95 ± 24.96			25.30 ± 26.41			
**COPM-P-T**	**T1**	8.60 ± 0.84	3.64	4.72	5.45 ± 2.28	1.46	0.80	**3.34/0.040 ***
	**T2**	10.26 ± 0.91			5.80 ± 2.48			
	**T3**	10.78 ± 0.99			5.70 ± 3.22			
**COPM-M-T**	**T1**	8.60 ± 4.05	0.75	2.87	5.55 ± 2.41	0.16	0.35	**6.68/0.006 ***
	**T2**	10.26 ± 4.39			5.15 ± 2.47			
	**T3**	10.78 ± 4.76			5.45 ± 3.21			
**GAS**	**T1**	36.36 ± 0.04	4.30	4.64	36.34 ± 0.05	0.57	0.50	**134.57/0.000 ***
	**T2**	62.28 ± 8.52			37.94 ± 3.96			
	**T3**	60.14 ± 8.82			37.94 ± 2.78			

TR-SHP, Telerehabilitation-Based Structured Home Program; UC, usual care; X, mean; SD, standard deviation; GAS, Goal Attainment Scale; GMFM-66, Gross Motor Function Measure-66; COPM-P, Canadian Occupational Performance Measure Performance; COPM-S, Canadian Occupational Performance Measure Satisfaction; PEDI-SC, Pediatric Evaluation of Disability Inventory Self Care; PEDI-MOB, Pediatric Evaluation of Disability Inventory Mobilization; PEDI-SF, Pediatric Evaluation of Disability Inventory Social Function; T, testing session; * two-way repeated measures mixed ANOVA; * *p* < 0.05.

**Table 6 children-10-00424-t006:** Home program implementation dose comparison of groups.

	UC + TR-SHP (*n* = 23)	UC (*n* = 20)	Z	*p* *
	Mean ± SD	Mean ± SD		
HP dose (hour)	55.95 ± 0.20	27.70 ± 15.53	−5530	0.000

HP, home program; UC, usual care; SD, standard deviation; Z, Mann–Whitney U test; * *p* < 0.05.

## Data Availability

The dataset analyzed in this study can be requested from Sinem Asena Sel (sinem.sel4@gmail.com) on reasonable request.

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
