# Peer review of "Effects of Telerehabilitation-Based Structured Home Program on Activity, Participation and Goal Achievement in Preschool Children with Cerebral Palsy: A Triple-Blinded Randomized Controlled Trial"

_children, 2023, doi:10.3390/children10030424_

Round 1

Reviewer 1 Report

This study is evaluating the effects of adding a weekly telerehabilitation session to standard care. Telerehabilitation became very important during the pandemic, and it is great to see a trial investigating efficacy to help determine whether such programs have value outside of lockdown periods.

- Some additional detail is needed in the Methods section - a sentence or two regarding study design eg parallel group, single blind (if that is true?) study, what is the primary aim that sample size was designed to investigate vs secondary aims that the study may or may not be appropriately powered to investigate.

- The blinding process isn't clear: p4 line 122 says that the first researcher [who administered the GMFM] was completely blinded from the scoring and statistical process, but it is more important that the researchers who are scoring and analysing the data and blinded in order to reduce bias.

- P5 line 185 describes home program content check. How was it checked, by reading family diaries? Or through video review? 

- P6 line 199 says that the control group was controlled by PT for 6 months. How were they controlled (or what does controlled mean)?

- Figure 3 is not very explanatory, it currently implies that the PT works with the family and child, but that the family and child are separate from the goal setting, home program education and program. It was useful to see the order of events/phases though. Perhaps it can be improved? For example by showing which aspects/phases are the same between the two study groups who operate in parallel?

- I am very confused about the GAS baseline data. What do the group means indicate for a goal-setting tool which, by definition, starts at zero (ie, there has been no attempt at attaining a goal yet)? Likewise COPM. GAS data is used very differently in my experience, so please give more information in methodology and results. 

- Are data in tables 2 and 3 meant to match up? Table 2 presents baseline data, Table 3 presents baseline data and follow up data, yet the baseline data is different.

- Please define X in the footnotes to Tables 3 and 4 (all other definitions are really clear thanks)

- If I have understood correctly, a primary goal of the TR counseling sessions is to increase the amount of quality of home therapy? Therefore, please present home program data for the two groups. I assume the TR group did significantly more home therapy than the control group, an important mediator in functional/activity/participation outcomes? 

It would be great to see more in the discussion:

- if children receiving usual care gained no benefit at all, it is pointless? A poor use of resources?!

- Family quality of life was not examined. Is it possible that this TR program puts a significant burden on the family?

- Discussion of confounders, eg the groups were too small to achieve comparability (there was a trend to greater severity CP in the control group) which is always difficult in a disorder like CP that is very heterogeneous. Alternatively, this could be added to the Limitations section along with the fact that families can't be blinded.

Also, I can't tell whether the TR group did better simply because they had more contact time with a therapist than the usual care group, or because the TR therapist knew the family goals (p6 line 193 implies that goals are set by external therapists without engaging the family or child during usual care, and that the usual-care-therapist may not be informed of family- and child-set goals developed with the researcher). Does it matter if the rehab is remote or not? 

Reviewer 2 Report

1. What was the education level of the parents and how could you ensure understanding of the treatment by telehealth .

2. Could a young child actively actively participate remote with short attention span and limited cognition?

3.Please comment if you would expect carry over after 12 weeks or attrition over time.

4. Provide details about communication levels of child and cognitive level of each child and relate to outcome on all scales.

5. Any differences in outcome if mother was the primary care giver or father?

6. Updated references.
